# Can Novel Synthetic Disperse Dyes for Polyester Fabric Dyeing Provide Added Value?

**DOI:** 10.3390/polym15081845

**Published:** 2023-04-11

**Authors:** Alya M. Al-Etaibi, Morsy Ahmed El-Apasery

**Affiliations:** 1Natural Science Department, College of Health Science, Public Authority for Applied Education and Training, Fayha 72853, Kuwait; 2Dyeing, Printing and Textile Auxiliaries Department, Textile Research and Technology Institute, National Research Centre, 33 El Buhouth St., Dokki, Cairo 12622, Egypt

**Keywords:** polyester fabrics, disperse dyes, added value, ultraviolet protection factor, biological activities

## Abstract

In this review, we present preparation methods for a series of new disperse dyes that we have synthesized over the past thirteen years in an environmentally safe and economical way using innovative methods, conventional methods, or using microwave technology as a safe and uniform method of heating. The results showed that in many of the synthetic reactions we carried out, the use of the microwave strategy provides us with the product in minutes and with higher productivity compared to the conventional methods. This strategy provides or may dispense with the use of harmful organic solvents. As an environmentally friendly approach, we used microwave technology in dyeing polyester fabrics at 130 degrees Celsius, and then, we also introduced ultrasound technology in dyeing polyester fabrics at 80 degrees Celsius as an alternative to dyeing methods at the boiling point of water. Here, the goal was not only to save energy, but also to obtain a color depth higher than the color depth that can be obtained by traditional dyeing methods. It is worth noting that obtaining a higher color depth and using less energy means that the amount of dye remaining in the dyeing bath is less, which facilitates the processing of dyeing baths and therefore does not cause harm to the environment. It is necessary after obtaining dyed polyester fabrics to show their fastness properties, so we explained that these dyes have high fastness properties. The next thought was to use nano-metal oxides to treat polyester fabrics in order to provide these fabrics with important properties. Therefore, we present the strategy for treating polyester fabrics with titanium dioxide nano-particles (TiO_2_ NPs) or zinc oxide nano-particles (ZnO NPs) in order to enhance their anti-microbial properties, increase their UV protection, increase their light fastness, and enhance their self-cleaning properties. We reviewed the biological activity of all of the newly prepared dyes and showed that most of these dyes possess strong biological activity.

## 1. Introduction

Since the beginning of creation, natural colors have played a special role in people’s lives, as they saw the Earth covered with red or yellow, the blue sky, and roses of many and varied colors. Disperse dyes are among the most widely used synthetic dyes used in the past two decades because of their ease of synthesis, attractive bright colors, and their wide use in polyester dyeing and printing [1,2]. Polyester (PET) is the most water-resistant fiber. This fiber is less prone to wrinkling and has excellent washability. The polyester dyeing process is carried out by using disperse dyes at higher temperatures and pressures. We have recently utilized ultrasound energy to improve the dyeing of polyester fabrics when dyeing with disperse dyes [3]. One of the environmentally friendly advantages of using microwave energy is rapid heating to high temperatures allowing greater ease of reactions, because the increased frequency of molecular vibrations during microwave irradiation speeds up these reactions [4,5,6,7,8,9,10,11,12,13,14]. Enaminones, pyridone, and pyrazolopyrimidines are key intermediates in the preparation of dispersion dyes due to their low cost and excellent antimicrobial activity. We know very well that fabrics can act as carriers or ideal environments for the growth of germs. Therefore, it is necessary to provide anti-microbial fabrics, so we had to improve the functionality of polyester fabrics using nanotechnology to increase their antibacterial and UV protection properties. It is well known that skin cancer results from prolonged exposure to ultraviolet radiation from the sun. Therefore, choosing a fabric designed and made of UV-blocking materials is automatically the right choice. In order to improve the functional performance of polyester fabrics, nano-metal oxides of titanium dioxide and zinc oxide were used.In this review article, we present the contributions of our laboratories over the past thirteen years using modern techniques such as microwave technology [15,16,17,18,19,20,21,22,23,24,25,26,27,28,29,30,31,32,33,34,35,36,37,38,39,40,41,42,43,44,45,46,47,48,49,50,51,52,53,54,55,56,57,58,59,60] or ultrasound technology to synthesize several new disperse dyes while demonstrating their biological activities. We will not only introduce the use of modern technologies such as ultrasound or microwave technology in dyeing polyester fabrics and compare them with traditional dyeing methods, but we will also discuss the process of treating polyester fabrics dyed with TiO_2_ NPs or ZnO NPs, and whether the treatment process with nano-metal oxides achieves a factor of protection from ultraviolet radiation. In addition to that, the treatment process gives polyester fabrics the ability to improve both their self-cleaning properties, the properties of light stability, and the properties of antimicrobial activities. Finally, in this review, we discuss whether the new dispersive dyes that we have synthesized have added value or not.

## 2. Materials

Microwave technology or ultrasound technology has many advantages. The beginning of this research team was in 2011 whenwe aimed to synthesize new disperse dyes using microwave heating by reacting hydrazonocyanoacetate with hydrazine hydrate [43]. As a result, dye **3** was created upon the reaction of hydrazonocyanoacetate 1 and hydrazine hydrate 2. We reported that compound **3** rapidly condensed with acetylacetone **4** to form the dispersed dye **5** using the microwave technique (Figure 1, Figure 1). The reaction of compound **3** with enaminones **6a–d** through microwave irradiation to synthesize the disperse dyes **7a–d** can be seen in (Figure 1).

Hydrazine hydrate **2** was refluxed with compound 8 to produce dispersion dye **10** [44,45]. Disperse dye **11** was made by combining hydrazone **9** and hydrazine hydrate and refluxing the mixture for four hours while adding ethanol as a solvent. NOE (nuclear Overhauser effects) difference measurements show that irradiating the compound 11-corresponding NH signal at 11.94 ppm increased the methyl proton signal at 2.36 ppm. Dye **10** and dye **11** easily condensed with the enaminones **6a–d** through refluxing for an hour in the presence of acetic acid and sodium acetate to produce disperse dyes **12a–d** or **12f–h** [45] (Figure 2).

The interaction between diazonium chloride and pyridones was used as the initial reaction technique for the manufacture of these azo colors. In 2014 [46], using microwave irradiation at 160 °C for 20 min, we disclosed a three-component condensation of ethyl cyanoacetates **13**, ethyl amines 14, and methyl propionylacetate **15** as ketoesters to produce pyridine-diones **16**. It is crucial to remember that in 2013 [42], we produced compound **16** by conventionally heating it for 6 h (Figure 3). The disperse dyes **18a–i** that exist in the hydrazone tautomeric state can be produced by combining pyridine **16** and various diazonium salts **17** based on X-ray crystallographic structure determination, as indicated in Figure 3 (Figure 2, Figure 3 and Figure 4).

Numerous researchers have examined aminothiophene derivatives as azo-disperse dyes in dyeing synthetic fibersdue to the importance of thiophene molecules and their high biological activity. Despite numerous investigations on the usefulness of these compounds in dye manufacturing, we are aware of no reports of the corresponding arylazothiopyridazines as promising monochromatic disperse dyes.

By reacting 7-Amino-4-benzotriazol-1-yl-2-p-tolyl-2H-thieno[3,4-d]pyridazin-1-one **19** with various diazonium salts **17**, we were able to create certain arylazothienopyridazines disperse dyes **20a–d** in 2014. This process was simple and eco-friendly. The interest in the synthesis of arylazothienopyridazines (Figure 3) was maintained by this work [60].

We were able to create unique, environmentally friendly disperse dyes in 2015 [57,58,59] (Figure 3). In order to do this, we combined the phenyl diazonium salt **17a** with the enaminones **6e** and **6f** in an acidic solution to create the mono azo disperse dyes **21a** and **21b**, which were 3-oxo-2-(phenylhydrazono)-3-p-arylpropionaldehydes.

## 3. Dyeing

When polyester fabrics were dyed using disperse dyes **3**, **5**, **7a–d**, **10**, **11**, and **18a–i** using microwave heating at 130 °C, a variety of multi-hued colors were produced (Table 1). The Kubelka–Munk equation was used to assess color intensity K/S.
$$K/S=\frac{{\left(1-R\right)}^{2}}{2R}-\frac{{\left(1-R₀\right)}^{2}}{2R₀}$$
where R is the decimal fraction of the reflectance of the dyed fabric; R_0_ is the decimal fraction of the reflectance of the non-dyed fabric; K is the absorption coefficient; and S is the scattering coefficient.

We discovered that polyester materials respond quite well to our new disperse dyes, giving them vivid, striking colors. However, we discovered that with some dyes, reusing dyeing baths is an easy, affordable, and economical way to treat the dyeing baths as well as obtain dyed fabrics for free. As a result, we chose to repurpose the dyeing bath and increase the dyeing time from 60 to 90 min without adding any additional dye.

According to Table 1, dye **12d** produces substantially stronger colors than dyes **12a** through **12c** and **12h**. At high pressure and high temperature, compounds **18a–i**, **20a–d**, **21a**, and **21b** were used to dye polyester fabrics, resulting in polyester fabrics with a range of colors from yellow to violet. *L** stands for lightness and (C) stands for chroma in the CIELAB (Color space) psychometric coordinates, which were developed by the International Committee on Illumination (CIE) in 1976. Table 2’s data demonstrate that when the dye’s hue was expressed as (*h*) values, practically all of the colored polyester fabrics conveyed a comparable hue. The dyed polyester materials’ color hues moved in a reddish direction, according to the positive estimates of *(b**) [46]. The UltraScan Pro (Hunter Lab, USA) 10° observer with D65 illuminant, d/2 viewing geometry, and a measurement area of 2 mm was used to measure the total color difference, *ΔE**. The following equation was used to determine the overall color difference between the sample and the standard, denoted as ∆*E* *:$$∆E^{*}=\sqrt{{\left(∆L*\right)}^{2}+{\left(∆a*\right)}^{2}{+(∆b*)}^{2}}$$
where ∆*L**, ∆*a**, and ∆*b** are the derivatives of corresponding parameters

Based on the colorimetric analysis, we can observe that the colors of polyester fabrics dyed with dyes **18e–g** (the values of *L* = 82.00, 81.222, and 79.46) were lighter and brighter than the colors of polyester fabrics dyed with dyes **18b–d** (the values of *L* = 59.35, 67.77, and 74.91) because the benzene ring contained electron donating groups, which reduced brightness, whereas the benzene ring contained electron withdrawing groups, which improved illumination and brightness.

### Dye Uptake

Table 2 lists the color strength values for polyester fabrics dyed at high and low temperatures and demonstrates that the high-temperature dyed fabrics were darker than the low-temperature dyed fabrics. The K/S values for dyes **18h** and **18i** for polyester fabric for dyeing at high and low temperatures were 19.38, 12.63, 4.74, and 3.46. According to these findings, the color strength of fabrics dyed at high temperatures was 309% and 265% more than the color strength of fabrics dyed at low temperatures. In addition, for fabrics dyed at high and low temperatures, the K/S values for dyes **21a** and **21b** were 17.59, 16.69, 12.21, and 8.97. These findings demonstrated that the color strength of the fabrics dyed at high temperatures were 144% and 186% higher than those of the color strength at low temperatures. Because it lessens pollution from dye waste, which has a substantially detrimental impact on the environment, high-temperature dyeing is a better ecologically friendly method, according to the information presented above. In addition, it is possible to suppose that while dyeing fabrics at high temperatures, the kinetic energy of the dye molecules can be increased by the temperature and may cause polyester fabrics to swell, which results in a rise in the dyeing rate compared to low-temperature dyeing. The results listed in Table 2 revealed that utilizing ultrasonic waves for conducting dyeing processes is superior to using the traditional approach since the value of K/S for ultrasonic dyeing at 80°C is 9.07; while it was 4.47 for dyeing by the conventional way at 100°C for disperse dye **18h**.

## 4. Fastness Properties

The fastness properties of the dyed samples against perspiration, rubbing, washing, and light were evaluated according to the American Society of Textile Chemists and Colorists tests. Table 3’s findings demonstrate that it was possible to measure the color fastness characteristics of textiles made of polyester dyed with dyes **3**, **5**, **7a–d**, **10**, **11**, **12a–h**, **18a–i**, **20a–d**, and **21a–b**.

## 5. Antioxidant Activity

We utilized an in vitro assay to determine the two dispersive dyes’ antioxidant capacities and their capacity to scavenge DPPH free radicals. The antioxidant activity of the dyes was quantified using their IC_50_ values. According to the findings, disperse dye **18h** had weak antioxidant activity (IC_50_) with a value of 191.6 and good antioxidant activity (IC_50_) with a value of 64.5 for disperse dye **18i** (Figure 5).

## 6. In Vitro Cytotoxicity Screening

One of the most crucial biological assessments is cytotoxicity assessment since there are various in vitro methods of cytotoxicity of drugs [48], such as the inhibition of protein synthesis or the inhibition of permanent binding to receptors. Four human cell lines, including HepG-2 cells (for the treatment of hepatocellular carcinoma), MCF-7 cells (for the treatment of breast cancer), HCT-116 cells (for the treatment of colon cancer), and A-549 cells (for the treatment of lung cancer), were used to investigate the initial anticancer activity of the synthetic dyes **18h** and **18i**. Using various concentrations of the disperse dyes, the values of IC_50_—the concentration required to stop 50% of the development of the culture when cells are exposed to the tested disperse dyes for 48 h—were computed. Table 4 and Figure 6 and Figure 7 demonstrate the dye’s substantial activity for dispense dye **18 h**, with IC_50_ values of 23.4, 62.2, 28, and 53.6 g/mL in the HePG-2, MCF-7, HCT-116, and A-549 cells, respectively. The IC_50_ values for disperse dye **18i**, on the other hand, were 196, 482, 242, and 456 g/mL in HePG-2, MCF-7, HCT-116, and A-549 cells, respectively.

## 7. Antimicrobial Activities of the New Synthesized Disperse Dyes

The agar diffusion method was employed to examine the antibacterial effects of the newly synthesized dyes **1**, **3**, **5**, **7a–d**, **18a–i**, and **21a,b** against bacteria and yeast while examining the bactericidal effects of the new disperse dyes we produced. The findings in Table 5 demonstrate strong antibacterial promoter activity. Disperse dyes **1** and **3** had considerable antibacterial activity against Gram-positive bacteria, in contrast to the other disperse dyes, which displayed only moderate to poor antibacterial capabilities.

It should be noted that Figure 8, Figure 9 and Figure 10 illustrates how dyes No. **1**, **3**, **5**, and **7a** affect the cellular states of *Bacillus subtilus*, *Staphylococcus aureus*, and *Candidia albicans*. The inhibitory zone remained unchanged after a day of incubation.

Table 5 demonstrates that all of the dispersion dyes tested demonstrated high positive antibacterial activity against pathogens based on the results for the inhibition zone diameter of the dyes **18a–a**, **21a**, and **21b**. We can say that these novel colors have inhibitory activity for many bacteria and fungi from the bacteria and fungi that were studied and can be employed for various pharmacological and medical objectives.

## 8. Treatment of Polyester Fabrics with TiO_2_ NPs or ZnO NPs

### 8.1. Antimicrobial Activity of Untreated and Treated Polyester Fabrics with TiO_2_ NPs or ZnO NPs

#### 8.1.1. Antimicrobial Activity of Untreated Polyester Fabrics

According to the antimicrobial examination findings presented in Table 6 and Figure 11
*Aspergillus flavus* and *Penicillium chrysogenum* are two forms of pathogenic fungi that the untreated polyester fabrics dyed with disperse dyes **18h** and **18i** under evaluation did not exhibit any antibacterial activities against *Aspergillus flavus* and *Penicillium chrysogenum*. While untreated polyester fabric dyed with disperse dye **18i** has strong antibacterial properties against *Pseudomonas aeruginosa* and very strong antibacterial properties against *Escherichia coli* (Figure 12), untreated polyester fabric dyed with the dye **18h** does not have antibacterial properties against all bacterial strains under study(Figure 11).

Also investigated was the antibacterial effectiveness of colored polyester fabrics against specific microbes. Untreated polyester fabrics against *Bacillus subtilis*, *Staphylococcus aureus* (Gram-positive), *Escherichia coli*, *Klebsiella pneumoniae* (Gram-negative), and *Candida albicans* (yeast) were used in the study under investigations. Table 7 clearly reveals that the colored polyester fabrics with dye disperse dyes **21a** or **21b** did not exhibit antibacterial action against all of the investigational microorganisms.

#### 8.1.2. Antimicrobial Activity of Treated Polyester Tabrics with TiO_2_ NPs or ZnO NPs

According to the antifungal screening results shown in Table 8, polyester fabrics dyed with disperse **18i** and treated with TiO_2_ NPs did not have any antifungal capabilities against the harmful fungi *Aspergillus flavus* and *Penicillium chrysogenum*, whereas polyester fabrics treated with TiO_2_ NPs and dyed with disperse **18h** have antifungal properties of the same two types.

Since the formation of active oxygen species such as hydrogen peroxide, superoxide anions, hydroxyl radicals, and single oxygen results in the destruction of the bacterial cell, we can say that the catalytic effect of TiO_2_ NPs and metal oxide nanoparticles is the primary cause of their antimicrobial effect.

In addition, although polyester fabrics colored with disperse dye **21a** and treated with ZnO NPs showed antibacterial action solely against *Bacillus subtilis*, polyester fabrics dyed with disperse dye **21b** and treated with ZnO NPs demonstrated antibacterial activity against both *Bacillus subtilis* and *Klebsiella pneumoniae*. We can also suppose that nano ZnO has antibacterial and antibacterial activity, and probably the reason is that ZnO nanoparticles may disrupt the bacterial membrane and inhibit their growth. Alternatively, nano ZnO may cause the formation of peroxide, which may have antibacterial properties.

### 8.2. UV Protective Properties of Untreated and Treated Polyester Fabrics with ZnO NPs or TiO_2_ NPs

First of all, we should be aware that UPF refers to a fabric’s ability to block UV rays. For its UV protection qualities, the ultraviolet protection factor (UPF) has been calculated. The UV blocking information for polyester fabrics treated with TiO_2_ or ZnO nano-particles is shown in Table 9. With values of 236.2 for dye 18b and 25.5 for disperse dye **18i**, Table 9 demonstrates that dyed polyester fabrics have greater UPF values than the non-dyed polyester fabrics. The dyed polyester fabrics treated with TiO_2_ nano-particles had UPF values of 283.60 for disperse dyes **18b** and 34.9 for disperse dye **18i**, according to Table 6’s UPF values.

This demonstrates that the UPF values of the polyester textiles treated with TiO_2_ particles and dyed with disperse dye **18i** are greater than those of the polyester fabrics treated with TiO_2_ particles and dyed with disperse dye **18i**. As a result, Table 9 demonstrates that for polyester fabrics that have been dyed had greater UPF values than polyester fabrics that have not been dyed, the respective values were 141.88 for disperse dye **21a** and 122.37 for disperse **dye 21b**. The dyed polyester fabrics treated with ZnO nanoparticles had UPF values of 173.25 for disperse dye **21a** and 190.59 for disperse dye **21b**, according to the UPF values in Table 9. The polyester fabrics treated with ZnO nanoparticles and dyed with disperse dye **21b** clearly have lower UPF values than the polyester fabrics treated with ZnO NPs and dyed with disperse dye **21a**.

### 8.3. Light Fastness of Untreated and Treated Polyester Fabrics with TiO_2_ NPs or ZnO NPs

With the use of disperse dyes **18h**, **18i**, **21a**, and **21b**, the light fastness characteristics of the dyed polyester fabrics treated with TiO_2_ NPs or ZnO NPs nanoparticles were investigated. The results were excellent and encouraging. With the exception of disperse dye **21a**, Table 9 demonstrates that treatment with TiO_2_ NPs or ZnO NPs nanoparticles more successfully demonstrates that the treated polyester fabrics have higher light fastness than the untreated samples (Table 9).

### 8.4. Self-Cleaning of Untreated and Treated Polyester Fabrics with TiO_2_ NPs or ZnO NPs

One advantage of polyester fabrics treated with nano-particles is that they change absorbed light into compounds that can clean themselves and remove stains. The photolysis and photolysis of methylene blue or methylene red adsorbed on polyester fabrics treated with TiO_2_ or ZnO nano-particles were researched to obtain the self-cleaning properties of TiO_2_ or ZnO nano-particles. Table 9 displays the effects of methylene blue or methyl red stains on polyester fabrics treated with TiO_2_ or ZnO nano-particles after 12 or 24 h of UV exposure. Methylene blue or methylene red stains exposed to UV radiation on polyester fabrics treated with TiO_2_ or ZnO nano-particles showed a partial transformation. The results demonstrated that the maximum rates of photolysis on the surface were between 60 and 80 percent after 12 h for the methylene blue spots treated with TiO_2_ NPs and between 60 and 70 percent after 24 h for the methylene red spots treated with ZnO NPs (Table 9). These positive results may be attributable to the fact that when the polyester fabric is treated with TiO_2_ NPs or ZnO NPs, thin layers of these particles form, which causes the fabric’s water-repellent properties to wrinkle. The waterproof surface keeps the polyester surface clean by preventing dirt from adhering.

At the end of this review, we have presented the methods of preparing a series of new disperse dyes in an environmentally safe way, using microwave technology or ultrasonic technology in dyeing polyester fabrics as an environmentally friendly approach. The new disperse dyes have an added value represented in their possession of the properties of high stability of the fabrics with which they were dyed and even the possession of these fabrics of anti-microbial properties and the property of protection from ultraviolet radiation. In addition, these new disperse dyes have an added value because most of them have strong biological activities.

## 9. Conclusions

We have shed light on the synthesis of a new series of novel disperse dyes by environmentally safe methods. We explained that these dyes have an added value represented in the fact thatthese dyes have great biological activity against Gram-negative and Gram-positive bacteria, as well as various fungi and yeasts that cause many diseases. In addition, polyester fabrics dyed with these dyes have biological activity, which qualifies these fabrics for use in many medical activities. The added value of these new disperse dyes was also discussed, not only in their expected use for dyeing polyester fabrics with high fastness properties, which reflects the importance of these disperse dyes, but we also presented methods of imparting various properties to polyester fabrics such as self-cleaning properties, maximizing light fastness, and maximizing antimicrobial activities when treating polyester fabrics using nano titanium dioxide or nano zinc oxide. Finally, we showed that these disperse dyes have an added value represented in that these disperse dyes have antioxidants and anticancer activities against some common cancers such as lung, breast, liver, and colon cancer.

## Data Availability

The data presented in this study are available on request from the corresponding author.

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
