# Peer review of "Can Novel Synthetic Disperse Dyes for Polyester Fabric Dyeing Provide Added Value?"

_polymers, 2023, doi:10.3390/polym15081845_

Round 1

Reviewer 1 Report

First of all, thank you very much for choosing our journal for your article. It is an area that has not been mentioned much until now. That's why I found it very valuable that you chose this field. A very well written and meticulous article. If you provide the corrections I mentioned, the article is suitable for publication for me.

- There's a huge gap after line 84, please make it look neater.

- Pages 5, 6 and 7 have consecutive schemes and figures. Please try to distribute them proportionally within the article.

- In line 146, please remove the space between 130 and the degree symbol.

Please note the same situation on line 151.

Table 1 has three undetected results. Could you give more information about the reasons for these?

- In line 231, please remove the space between Fig. and 5.

Another error of the same style appears on line 255.

Leave enough space between the text of table 5 on line 271 and the table.

- In line 298, after bacteria, you put too much space before G

Another case of the same kind is present on line 299.

- In line 377, the treated polyester fabrics have higher light fastness than the untreated samples. How it is possible? Can you please explain more ?

- In line 395, The results demonstrated that the maximum rates of photolysis on the surface were between 60 and 80 percent after 12 hours for the methylene blue spots treated with TiO2 NPs, and between 60 and 70 percent after 24 hours for the methylene red spots treated with ZnO NPs. Do you have any idea about the process after 24 hours? For example, do you have available data on 36 hours?

Author Response

Please find attached our  response file

Reviewer 2 Report

The work presents a series of synthesised dyes that are used to dye polyester fabrics with environmentally friendly technologies. All the parameters analysed after using these technologies, reveal that the technology is effective. 

The work is very careful and meticulous and addresses one of the areas that can be most harmful to the environment, at the stage of dyeing fabrics.

Author Response

Please find attached our response file

Reviewer 3 Report

Dear authors

Thanks for your interesting paper, a lot of useful information. The different sections of the paper are well organized. There are few things that need to be handled. Please avoid writing too long sentences as it is difficult to follow. I highlighted for your convenience some parts in the document and resumed here my comments.

Regards

The authors need to revise the Abstract – few sentences (highlighted in the uploaded manuscript) are too long and difficult to read. Too many information are discussed in the same sentences and often repeated.

Page 2 line48 – the use of better here is confusing? Better respect to what? You need to reformulate the sentence. Line 49 – reports are limitative, the cited reference is not a technical note. Please change with Research or paper

Page 2 line 70 add a full stop. …protection from ultraviolet. In addition, the treatment process…

Page 4 line 94 you need to clarify the acronym NOE

Page 4 line 92 and 109 please move the figure before the full stop – no repetition.

Page 9 line 22 the affirmation should be based on evident data. As you performed colorimetric analysis you should say the value of L and not generically say lighter or brighter. You need to add the Lab value and correlated differences.

Page 10 line 204 please implement the Table caption – you need to say that colorimetric data are also present 

Author Response

Please find attached our response file
